# Involvement of *Laccase2* in Cuticle Sclerotization of the Whitefly, *Bemisia tabaci* Middle East–Asia Minor 1

**DOI:** 10.3390/insects13050471

**Published:** 2022-05-18

**Authors:** Chun-Hong Yang, Qi Zhang, Wan-Qing Zhu, Yan Shi, He-He Cao, Lei Guo, Dong Chu, Zhaozhi Lu, Tong-Xian Liu

**Affiliations:** Key Laboratory of Integrated Crop Pest Management of Shandong Province, College of Plant Health and Medicine, Qingdao Agricultural University, Qingdao 266109, China; chyang@qau.edu.cn (C.-H.Y.); qizhang@stu.qau.edu.cn (Q.Z.); wanqing.zhu@acrobiosystems.com (W.-Q.Z.); shiyanyuanyi@qau.edu.cn (Y.S.); caohehe1988@qau.edu.cn (H.-H.C.); guolei@qau.edu.cn (L.G.); chinachudong@sina.com (D.C.); zhaozhi_lv@sina.com (Z.L.)

**Keywords:** *Bemisia tabaci*, *Laccase2*, cuticle, expression profiling, RNA interference, insect pest control

## Abstract

**Simple Summary:**

*Bemisia tabaci* Middle East–Asia Minor I (MEAM1) is a widely distributed invasive agricultural pest that causes extensive damage to agricultural, horticultural, and ornamental crops. *Laccase2* (*Lac2*), a phenol oxidase, plays an important role in cuticle tanning of some insects. However, the function of *Lac2* in whitefly remains unclear. In this study, we identified a *BtLac2* gene in MEAM1 that is expressed in all developmental stages, and its expression in the cuticle is especially high. Knockdown of *BtLac2* in nymphs produced thinner and fragile cuticles, which significantly increased the mortality rate and extended the development duration of nymphs, as well as further decreasing the emergence rate of adults. Overall, *BtLac2* plays an important role in cuticle hardening of whitefly, suggesting a potential management strategy using RNAi to knock down *BtLac2* expression.

**Abstract:**

Cuticle sclerotization is critical for insect survival. *Laccase2* (*Lac2*) is a phenol oxidase that plays a key role in cuticle formation and pigmentation in a variety of insects. However, the function of *Lac2* in whitefly, *Bemisia tabaci*, remains unclear. In this study, we identified a *BtLac2* gene in *B. tabaci* MEAM1 and found that *BtLac2* was expressed in all stages. It was highly expressed in the egg stage, followed by nymph and adult. Moreover, the expression of *BtLac2* was higher in the cuticle than in other tissues. Knockdown of *BtLac2* in nymphs produced thinner and fragile cuticles, which significantly increased the mortality rate, extended the development duration of nymphs, and decreased the emergence rate of adults. This result demonstrates that *BtLac2* plays an important role in the cuticle hardening of *B. tabaci* and suggests a potential management strategy using RNAi to knock down *BtLac2* expression.

## 1. Introduction

Laccase (EC 1. 10. 3. 2) is a dioxygen oxidoreductase that contains four copper atoms in the catalytic center. Laccases are widely distributed in fungi, higher plants, insects, and some bacteria [1,2,3,4,5,6,7] and belong to the family of blue multi-copper oxidases (BMCOs) [8,9,10]. The wide range of substrates catalyzed by laccase contributes to the diversity of laccase function [11]. Previous studies have found that insect laccases mainly include *Laccase1* (*Lac1*) and *Laccase2* (*Lac2*); *Lac1* may play an important role in metal ion metabolism, detoxification of secondary plant compounds, and lignocellulose digestion [12,13,14,15], whereas *Lac2* plays an important role in insect cuticle tanning, morphology, and pigmentation [16,17,18,19].

*Lac2* is a highly conserved multicopper oxidase that has three classic Cu-oxidase domains with a Cu^2+^ in the catalytic center, and it is known that the enzyme is synthesized by the epithelial cells and is secreted to the locations where new epidermis synthesis occurs [17,20,21]. The red flour beetle *Tribolium castaneum* injected with dsRNA for *Lac2* showed soft cuticles, and the insects were deformed and subsequently died [16]. Similar phenotypes were also observed in other insects, such as *Monochamus alternatus* [17], *Apis mellifera* [18], and three hemipteran stinkbugs, *Riptortus pedestris* (Alydidae), *Nysius plebeius* (Lygaeidae), and *Megacopta punctatissima* (Plataspidae) [19]. In addition, many studies have indicated that *Lac2* also has other biological functions. For example, *Lac2* plays roles in *Vanessa cardui* butterfly wing pigmentation and scale development [22]. *Lac2* knockdown in *Aedes albopictus* resulted in pale and incomplete sclerotization of eggs and caused high mortality [23]. However, information is lacking on the specific role played by *Lac2* in whiteflies, which are important pests and vectors of plant viruses.

In this paper, we investigated the function of *Lac2* in *B. tabaci* cryptic species Middle East–Asia Minor I (MEAM1), also known as the B biotype, which is an invasive agricultural pest that is widely distributed and causes billions of dollars in crop losses worldwide [24,25,26,27]. In this study, we cloned the cDNA sequence of *BtLac2* of MEAM1. We characterized the temporal and spatial expression pattern in the developmental stages and body tissues of MEAM1 via real-time quantitative PCR (RT-qPCR). We also investigated the phenotype of MEAM1 after knockdown of *BtLac2* expression and its effect on the emergence and mortality of *B. tabaci*.

## 2. Materials and Methods

### 2.1. Whitefly Rearing

The MEAM1 whiteflies were maintained on tomato plants (*Lycopersicon esculentum*, cv. Zhongza 9) at 26 ± 1 °C, under a photoperiod of 16:8 h light/dark and 70 ± 10% RH. The purity of the whitefly colony was monitored by gene sequencing of *mtCOI* every two months.

### 2.2. Cloning, Sequence Retrieval, and Phylogenetic Analysis

Total RNA was extracted from approximately 200 *B. tabaci* adults using Trizol (Invitrogen, Carlsbad, CA, USA) and then reverse transcribed into cDNA using a PrimeScript^TM^ RT reagent Kit with gDNA Eraser (Takara, Bio Inc., Shiga, Japan). The *BtLac2* nucleotide sequence was found in the MEAM1 *B. tabaci* genome database (http://www.whiteflygenomics.org/cgi-bin/bta/index.cgi, accessed on 6 April 2019), and three pairs of primers were designed based on the putative *BtLac2* sequence to obtain the full open reading frame (ORF). PCR products were purified and subcloned into PMD-19 vector (Takara, Bio Inc., Shiga, Japan) for sequencing. The primers are listed in Table 1.

DNAMAN was used to translate the full-length sequence of the nucleotide ORF to an amino acid sequence. Transmembrane helices were predicted using the TMHMM server (http://www.cbs.dtu.dk/services/TMHMM, accessed on 7 May 2019). The SignalP server (http://www.cbs.dtu.dk/services/SignalP, accessed on 7 May 2019) was used to predict signal peptides. Alignment and phylogenetic analyses were performed using DNAMAN and MEGA 7.0 software, respectively.

### 2.3. RT-qPCR Analysis of BtLac2 Expression Patterns 

RNA was isolated from different developmental stages (eggs, 1st-instar nymph, 2nd-instar nymph, 3rd-instar nymph, 4th-instar nymph, and adults), and different tissues (head, thorax, cuticle, and ovary) of adults. Approximately 200 individuals of *B. tabaci* were used for one biological replicate. Adults that emerged within two days were used in the experiment. Subsequently, RT-qPCR was performed. The RT-qPCR primers (Table 1) were designed using Primer 3 (http://bioinfo.ut.ee/primer3-0.4.0/, accessed on 7 May 2019 ) and synthesized by Sangon Biotech (Shanghai, China). Succinate dehydrogenase complex subunit A (SDHA) was selected as a housekeeping gene for the RT-qPCR [28]. The RT-qPCR conditions were 95 °C for 30 s, followed by 40 cycles of 95 °C for 5 s and 60 °C for 20 s. The reactions were performed on a LightCycler^®^ 96 (Roche group, Basel, Switzerland). Three biological replicates, taken as three independent samples, were performed for each treatment.

### 2.4. Functional Analysis of the BtLac2 Gene in B. tabaci by RNAi

To explore the function of *BtLac2* in MEAM1, RNAi bioassay for the delivery of dsRNA by nanomaterial was performed. A unique region of the MEAM1 *BtLac2* gene was chosen as a template for double-stranded RNA (dsRNA) synthesis (Appendix A). The fragments of the *BtLac2* and enhanced green fluorescent protein (*EGFP*) genes were amplified by reverse transcription PCR (RT-PCR) using specific primers conjugated with 20 bases of the T7 RNA polymerase promoter (Table 1). The PCR products of *BtLac2* (478 bp) and *EGFP* (420 bp) were used as templates for dsRNA synthesis using the Transcript Aid T7 High Yield Transcription Kit (Thermo Scientifific, Vilnius, Lithuania). The dsRNA was ethano-precipitated, resuspended in nuclease-free water, quantified using a NanoPhotometer N50 (Implen, Munich, Germany), and stored at −20 °C until use. The concentration of the dsRNA solution was 0.5 µg/µL, as the solution contained 1 µg/µL dsRNA and an equal volume of nanomaterial provided by Professor Jie Sheng from China Agricultural University [29,30]. Twenty nanoliters of dsRNA mixture was dropped on the center of the back of a 4th-instar nymph using a micro-injector. The RNAi whiteflies were kept in a climate incubator (26 ± 1 °C, 60–70% RH, and 16L:8D photoperiod).

The RNAi efficiency was determined after 72 h. Thirty nymphs in each of the three replicates were used for total RNA extraction and RT-qPCR to determine the effect of RNAi. The RT-qPCRs were performed in a 20 µL mixture containing 1 µL of cDNA, 10 µL of 2× SYBR Premix Ex Taq II (Takara Biotechnology, Dalian, Liaoning, China), 1 µL of each primer (Table 1), and 7 µL of H_2_O. The RT-qPCR program and housekeeping gene used were the same as those described above. The enhanced green fluorescent protein (*EGFP*) gene was used as a control. The same-aged nymphs (13 days from egg stage) were used for RNAi.

Phenotypes were observed and captured 3 days, 8 days, and 10 days after dsRNA treatment using a stereomicroscope. Ten nymphs were used for phenotypic observation, and fifty nymphs for each condition were used for analysis of the emergence rates of RNAi and control; three biological replicates were performed. The numbers of emerged *B. tabaci* in both the control and treatment groups were counted every 24 h for a total of 10 days.

### 2.5. Cuticle Morphology Analysis

After 72 h of ds*BtLac2*/ds*EGFP* treatment, nymphs were prepared for histological sectioning and HE staining. The samples were fixed overnight in 4% (*w*/*v*) paraformaldehyde, dehydrated, embedded in paraffin, and sectioned. Slides were prepared by soaking twice in xylene for 20 min, twice in 100% alcohol for 5 min, and in 75% (*v*/*v*) alcohol for 5 min. The slides were then rinsed with water. The slides were immersed in hematoxylin solution for 3–5 min and rinsed with water. The sections were differentiated with acid alcohol, rinsed again, stained with ammonia solution, and washed in slowly running tap water. The sections were then stained in eosin using 85% (*v*/*v*) alcohol for 5 min, 95% (*v*/*v*) alcohol for 5 min, and eosin for 5 min. The sections were dehydrated thrice with 100% alcohol for 5 min, twice with xylene for 5 min, and, finally, in colorless hyaloid resin. Digital images of the sections were acquired using an Olympus fluorescence microscope (Olympus Optical, Tokyo, Japan) coupled to cellSens Standard software. The thickness of the dorsal section of the nymphs was measured using cellSens Standard software (Olympus Optical, Tokyo, Japan). 

### 2.6. Data Analysis

Different developmental stage and body tissue data were used to calculate the mean ± SEM and were analyzed by one-way analysis of variance (ANOVA) using SPSS 20.0 (IBM, Armonk, NY, USA). Student’s *t*-test was used to determine the significance of differences between the treatment and control groups. The figures were made using SigmaPlot 12.0 (Systat Software Inc., Palo Alto, CA, USA). The genes’ relative expression levels from RT-qPCR were calculated by the 2^−ΔΔCt^ method.

## 3. Results

### 3.1. Characterization of BtLac2 from B. tabaci

The complete cDNA of *BtLac2* contains an open reading frame (ORF) of 2238 bp encoding a protein of 745 amino acid residues with a predicted molecular weight of 82.53 kDa, and the theoretical isoelectric point (pI) is 6.31. GenBank detection revealed that *BtLac2* has high homology with *Lac2* of other insects and contains three Cu-oxidase domains (Appendix A), belonging to the laccase family. By using the biological software signalP, it was predicted that the signal peptide was located at amino acids 1–26, and we speculated that *BtLac2* is a secreted protein. There were 63 basic amino acids (Arg + Lys) and 73 acidic amino acids (Asp + Glu) in the sequence, accounting for 8.4% and 9.7% in the sequence, respectively. The instability coefficient was 36.73, suggesting that *BtLac2* is a stable protein.

The *Lac* family genes of different insects were found through NCBI’s BlastP, and a multiple amino acid sequence comparison of *Lac2* between *B. tabaci* cryptic species MEAM1 and other insects was performed to construct a phylogenetic tree of Lac using MEGA 7.0 (Figure 1A). Phylogenetic analysis based on the *Lac2* amino acid sequences showed that *BtLac2* is closely related to *Lac2* of *Nephotettix cincticeps, Megacopta punctatissima*, and *Riptortus pedestris,* sharing more than 78% homology. As a character of laccases, the well-conserved cysteine-rich region was found in the predicted *BtLac2* protein when it was aligned with laccase protein sequences from other insect species (Figure 1B). The *Lac2* gene is relatively distant from the *Lac*, *Lac1*, and *Lac4* genes of *B. tabaci*.

### 3.2. Expression Profiling of BtLac2

To understand *BtLac2* function, the expression levels of the *BtLac2* gene in different tissues of adults and all development stages were detected by RT-qPCR (Figure 2 and Figure 3). The results indicated that *BtLac2* mRNA was expressed consistently in all different developmental stages (eggs, 1st–4th-instar nymphs, and adults). However, the relative expression level of *BtLac2* in eggs was significantly higher than that in adults and nymphs. The expression level of the *BtLac2* gene in eggs was the highest, 130-fold higher than that in adults, which showed the lowest expression. In addition, the relative expression levels of *BtLac2* in 2nd- and 3rd-instar nymphs were significantly higher than that in adults (*p* < 0.05). The expression of *BtLac2* was detected in the head, thorax, cuticle, and ovary. It is worth noting that *BtLac2* was expressed the most in the cuticle. There was no significant difference between male and female individuals. 

### 3.3. Low Expression of BtLac2 Results in Nymph Cuticle Sclerotization Defects and Emergence Retardation

The thickness of the dorsal section of the ds*BtLac2*-treated nymphs (Figure 4B) was only about 37% (Figure 4C) of that in the control group (Figure 4A). The expression of the *BtLac2* gene was detected by RT-qPCR at 72 h post RNAi. It was shown that the *Lac2* gene can be successfully silenced by a nanocarrier-based cuticle penetration method, and the expression of the *Lac2* gene in the whitefly was decreased by 58% at 72 h (Figure 4D).

Meanwhile, compared to the control nymphs, ds*BtLac2*-treated nymphs were very fragile and easy to break (Figure 5(A1)); the nymphs could not develop and emerge as adults normally, and their bodies gradually dried out (Figure 5(C1)). The emergence rate of RNAi-treated nymphs (42 ± 2.49%) was significantly lower than that of control nymphs (78.67 ± 2.37 %) (*p* < 0.05) (Figure 6A), and the mortality rate of ds*BtLac2*-treated nymphs (0.21 ± 0.06) was significantly higher than that of control nymphs (0.58 ± 0.03) (*p* < 0.05) (Figure 6B). In addition, the development duration of the RNAi-treated nymphs was significantly increased (*p* < 0.05), and the time for emergence was nearly two days longer than that of the control nymphs (7.4 d ± 0.1 in control vs. 8.7 d ± 0.1 in treatment) (Figure 6A). Our data indicate that *BtLac2* dysfunction can lead to cuticle defects in the sclerotization processes, suggesting that it is crucial for cuticle tanning. The obvious time lag, low emergence rate, and physical weakness therefore affected the whitefly population. 

## 4. Discussion

Prior studies on *Lac2* genes in insects such as *T. castaneum*, *M. alternatus*, *A. mellifera*, *Bombyx mori*, and stinkbugs showed that *Lac2* is mainly involved in the hardening and darkening of the insect epidermis [16,17,18,19,30]. However, no studies had yet been performed to investigate the function of *Lac2* in the important agricultural pest *B. tabaci*. This is the first study to explore the important role of the *Lac2* gene in *B. tabaci*. 

In this work, we successfully cloned the CDS sequence of the *Lac2* gene from *B. tabaci* cryptic species MEAM1. *BtLac2* showed high similarity with *Lac2* from other insects at the amino acid level. Phylogenetic analysis based on *Lac2* amino acid sequences showed that *BtLac2* is closely related to *Lac2* of *N. cincticeps, M. punctatissima*, and *R. pedestris,* sharing more than 78% homology with all three proteins, indicating that *BtLac2* is related closely to laccase from these insect species. A comparison of the *Lac*, *Lac1, Lac2*, and *Lac4* genes of *B. tabaci* in the phylogenetic tree revealed that these genes are distinct from each other, suggesting that different laccases play different functions in insects. An analysis of the structural characteristics of the amino acid sequence showed that *BtLac2* has the typical laccase characteristic of Cu-oxidase domains. Copper ions are the active center of the laccase catalytic reaction and play an important role in the catalytic oxidation process. There are generally four Cu-oxidase domains in laccases from plants and fungi. However, similarly to those of other insects, the laccase of *B. tabaci* contains three Cu-oxidase domains [31,32], indicating that the function of *Lac2* in insects may differ from that in other organisms.

The RT-qPCR results indicated that *BtLac2* mRNA was expressed consistently in all developmental stages of the whitefly, suggesting that *BtLac2* is expressed throughout development and growth and plays a crucial role in biological processes. Higher expression levels of *BtLac2* were found in *B. tabaci* eggs and nymphs compared to adults. *Lac2* has been found in the eggshell [33,34], and pigmentation of the eggshell was blocked when the expression level of *Lac2* was lower [23]. Eggs and nymphs can protect themselves from external adverse factors mainly through passive defenses. A high expression level of *Lac2* in this stage can promote quinone accumulation to enhance, harden, and steady the cuticle [35]. Especially in the nymphal stage, *B. tabaci* must constantly molt and form a new epidermis during growth. The high expression level of *BtLac2* indicates that its function may play an important role in the formation of the epidermis of the whitefly nymphs. The RNAi study showed that silencing of *BtLac2* softened the whitefly nymphal epidermis, and the body gradually shriveled up, indicating that *BtLac2* is involved in the tanning process of the whitefly’s epidermis. It is also beneficial for adult cuticle formation to adapt to the environment [16,35]. *Laccase2* plays an important role in the cross-linking of cuticular proteins, a process that is involved in the oxidative conjugation of quinones and quinone methides [36]. Adaptation of the cuticular composition often leads to increased insecticide resistance due to inhibition of insecticide penetration capacity [33]. It has been demonstrated that *Lac2* contributes to multiple-resistance phenotypes [37].

Due to their limited feeding capacity, or maybe because they cannot transmit viruses, the egg and nymphal stages of whiteflies are considered less harmful to plants than the adult stage [38]. So, it is reasonable to disturb the normal development of whiteflies in their egg or nymphal stages to prevent further loss. In this study, a nanocarrier-based RNAi technology was used to silence the *Lac2* gene in the nymphal stage of *B. tabaci*. Nanomaterial could deliver dsRNA into cells, achieving a good silencing effect [27,29,39]. *Bemisia tabaci* with successful silencing of the *Lac2* gene showed thinner cuticles and fragile bodies, and they died due to their weak ability to resist the external environment. In addition, we also found that the nymphs in which the *BtLac2* gene was successfully interfered with took longer for emergence than the control group, and the emergence rate also decreased, indicating that *BtLac2* plays an important role in the development of *B. tabaci*. These results, along with those of previous research, enrich our knowledge about the functions of insect *Laccase2*, so that it may potentially act as a new target for pesticides and pest control in the future.

## 5. Conclusions

In this study, we cloned the cDNA of *BtLac2* from *B. tabaci* MEAM1. After gene expression analysis, we found that *BtLac2* was expressed in all developmental stages, but the highest expression level was observed in the egg stage, followed by nymphs and adults. Further, Bt*Lac2* was expressed more in the cuticle than in other tissues. Knockdown of *BtLac2* in nymphs produced thinner and fragile cuticles, which further impeded the emergence rate of adults, as well as their fitness. Our data could clarify the structures, phylogeny, expression pattern, and biological functions of *BtLac2* in *B. tabaci*.

## Figures and Tables

**Figure 1 insects-13-00471-f001:**
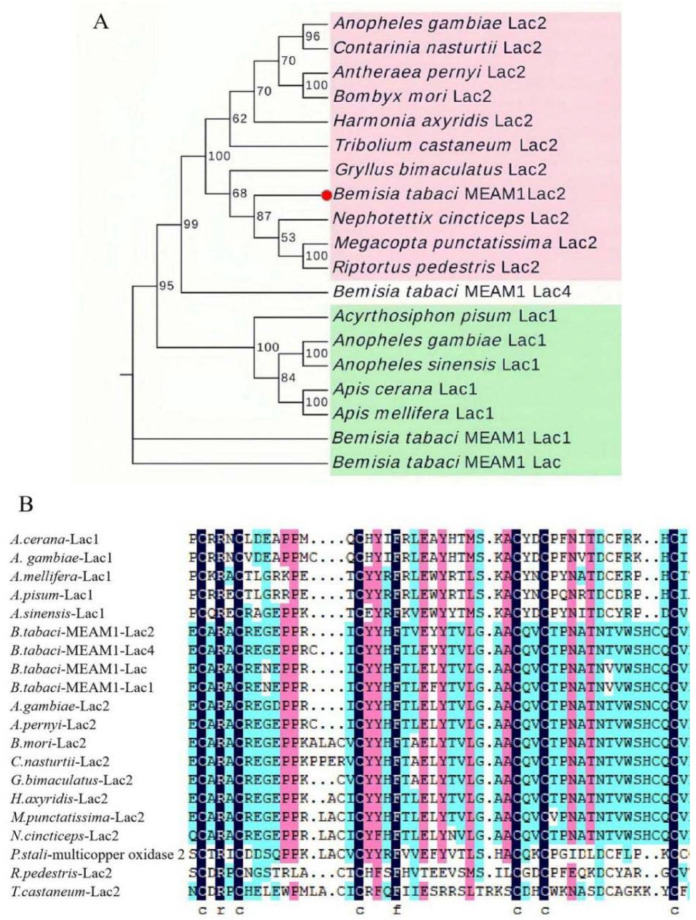
(**A**). Phylogenetic tree of Lac amino acid sequences in *B. tabaci* and other insects. The phylogenetic tree was generated by MEGA 7.0 based on the neighbor-joining method according to amino acid sequences. Accession numbers: *Apis cerana Lac1* XP_016918769.1; *Anopheles gambiae Lac1* AAN17505.1; *Apis mellifera Lac1* XP_001120790.2; *Acyrthosiphon pisum Lac1* XP_029344899.1; *Anopheles sinensis Lac1* KFB43437.1; *Bemisia tabaci* MEAM1 *Lac4* Bta00306; *Bemisia tabaci* MEAM1 Lac Bta11878; *Bemisia tabaci* MEAM1 *Lac1* Bta15284; *Anopheles gambiae Lac2* AAX49501.1; *Antheraea pernyi Lac2* AII19522.1; *Bombyx mori Lac2* BAG70891.1; *Contarinia nasturtii Lac2* XP_031632951.1; *Gryllus bimaculatus Lac2* BAM09185.1; *Harmonia axyridis Lac2* QNH91383.1; *Megacopta punctatissima Lac2* BAJ83488.1; *Nephotettix cincticeps Lac2* BAJ06133.1; *Plautia stali* multicopper oxidase 2 BBM95978.1; *Riptortus pedestris Lac2* BAJ83487.1; *Tribolium castaneum Lac2* AAX84203.2. (**B**). Multiple amino acid sequence alignment of the cysteine-rich consensus region in insect laccase proteins. Numbers on the right are the position of the final amino acid. Six cysteine residues conserved in the *Laccase1* and *Laccase2* proteins are marked by the letter C below the sequences.

**Figure 2 insects-13-00471-f002:**
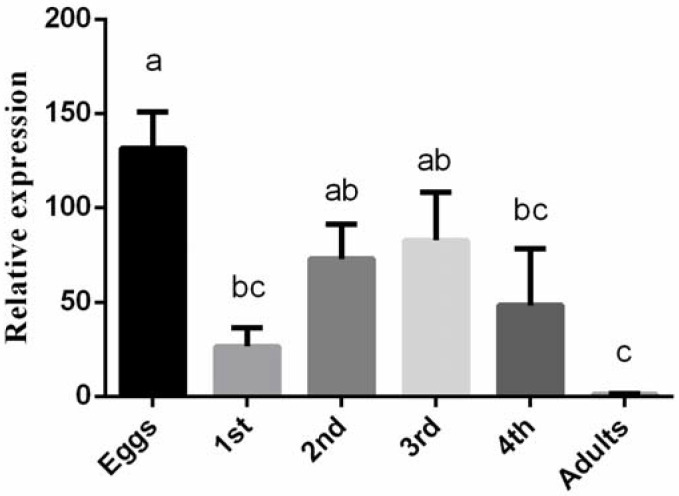
Relative expression levels of *BtLac2* in different developmental stages of *B. tabaci* cryptic species MEAM1. Bars indicate standard errors (*n* = 200–300, biological replicates = 3). Different letters indicate statistical significance (*p* < 0.05).

**Figure 3 insects-13-00471-f003:**
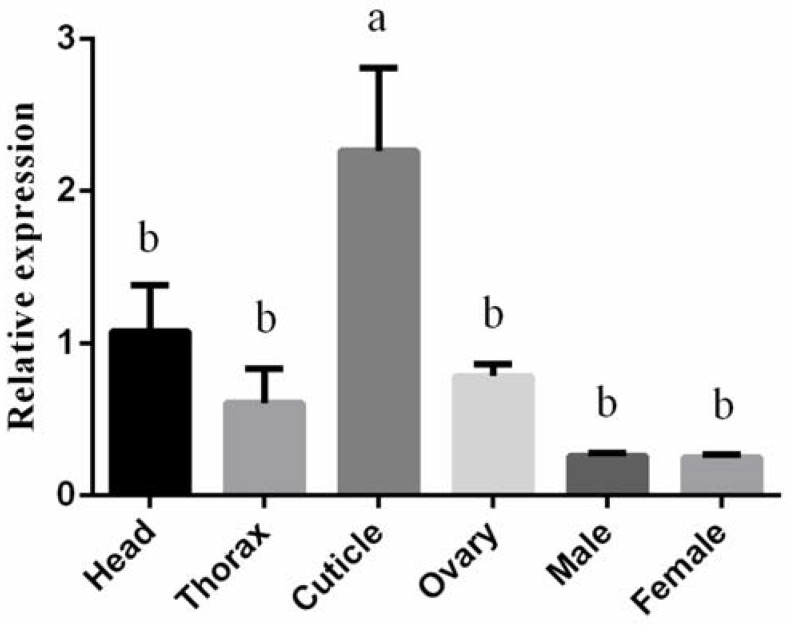
Relative expression levels of *BtLac2* in different tissues of *B. tabaci* cryptic species MEAM1. Bars indicate standard errors (*n* = 200–300, biological replicates = 3). Different letters indicate statistical significance (*p* < 0.05).

**Figure 4 insects-13-00471-f004:**
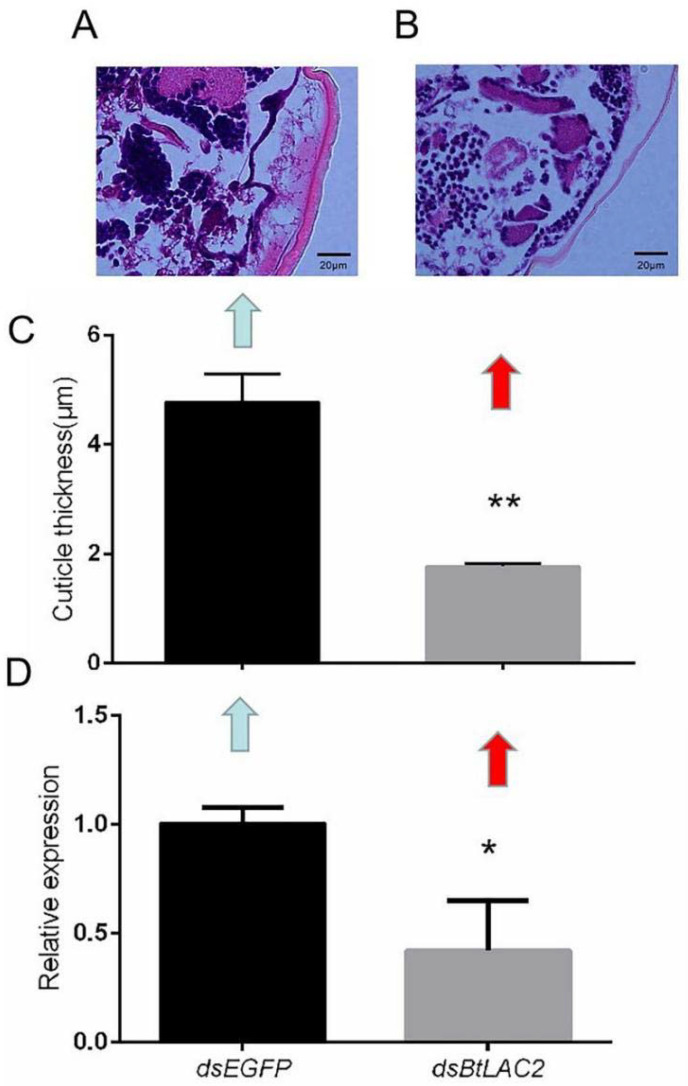
Effects of silencing *BtLac2* on physical characteristics of the *B. tabaci* nymph cuticle. Observation and thickness measurement of the dorsal cuticle of individual nymphs in the ds*EGFP*-treated (**A**) and ds*BtLac2*-treated (**B**) groups. (**C**) The dorsal cuticle thickness of *B. tabaci* 72 h after the RNA interference (*t* = 18.329, ** *p* < 0.01, *n* = 10). (**D**) The relative expression of the *BtLac2* gene 72 h after the RNA interference. Bars indicate standard errors (*t* = 3.863, *n* = 50, biological replicates = 3, * *p* < 0.05). dsRNA targeting enhanced green fluorescent protein (*EGFP*) was used as a negative control.

**Figure 5 insects-13-00471-f005:**
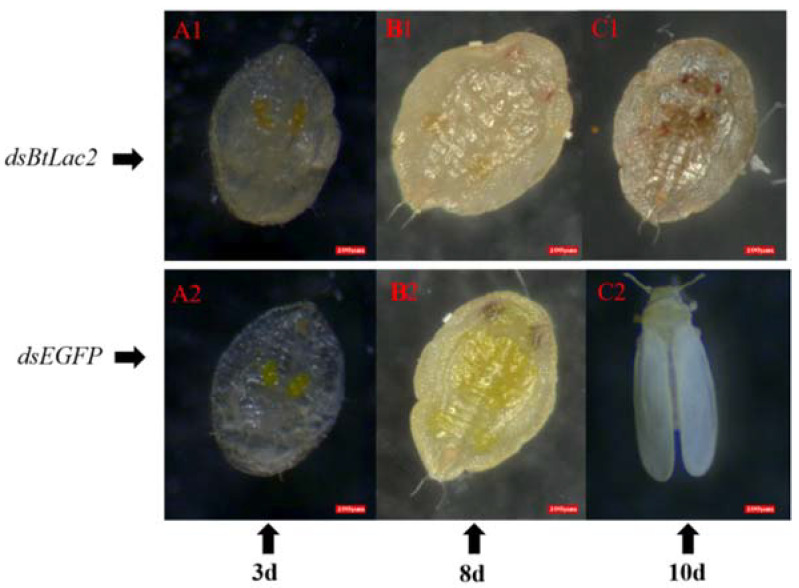
Lethal phenotypes of *B. tabaci* nymphs post RNA interference by *BtLac2*: after 3 days, 8 days, and 10 days. dsRNA targeting enhanced green fluorescent protein (*EGFP*) was used as a negative control.

**Figure 6 insects-13-00471-f006:**
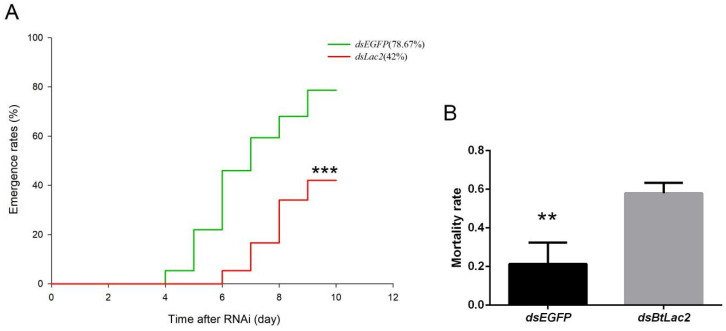
(**A**) The emergence rates of *B. tabaci* in the 10 days after knocking down of *Lac2*. The same-aged nymphs (13 days from egg stage) were used for the RNAi (*t* = 1.564, *n* = 50, biological replicates = 3, *** *p* < 0.001). (**B**) Mortality rate of *B. tabaci* nymphs 10 days after RNA interference by *BtLac2*. Bars indicate standard errors (*t* = 5.197, *n* = 50, biological replicates = 3, ** *p* < 0.01).

**Table 1 insects-13-00471-t001:** Primers Used in This Study.

Primer Name	Primer Sequence (5′→3′)	Amplicon (bp)	Remarks
*LAC2-1*	F:ATGAAGGTGAAAATGTCACG	777	fragment
	R:GTTTCCTTGTAAGATGGGGC		
*LAC2-2*	F:CCCATCTTACAAGGAAACACT	795	fragment
	R:CAGTTCTTGGAGTAAAGCCTT		
*LAC2-3*	F:AGGCTTTACTCCAAGAACTGC	704	fragment
	R:TCAATGTAAACTGACCGGG		
q-*LAC2*	F:AGTGTCTGCCCAGCTCAACT	213	qRT-PCR
	R:CAATTGCTGCACTCGTTTGT		
q-*SDHA*	F:GCGACTGATTCTTCTCCTGC	141	qRT-PCR
	R:TGGTGCCAACAGATTAGGTGC		
ds*BtLAC2*	F: taatacgactcactatagggGATCGACGACATCCCACCAG	479	RNAi
	R: taatacgactcactatagggATCGAGGGAAACCTTCTGGC		
ds*EGFP*	F: taatacgactcactatagggCACAAGTTCAGCGTGTCCG	420	RNAi
	R: taatacgactcactatagggTGCCGTTCTTCTGCTTGTCG		

## Data Availability

The data presented in this study are available on request from the corresponding author.

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
