# Peer review of "Involvement of Laccase2 in Cuticle Sclerotization of the Whitefly, Bemisia tabaci Middle East–Asia Minor 1"

_insects, 2022, doi:10.3390/insects13050471_

Round 1
Reviewer 1 Report
The manuscript “Involvement of Laccase 2 in cuticle sclerotization of the whitefly, Bemisia tabaci Middle East-Asia Minor 1” examined the expression and function of the phenol oxidase Laccase 2 in Bemisia tabaci. They found that Laccase 2 was highly expressed in the egg stage and cuticle. Knockdown of Laccase 2 resulted in thinner cuticle, increased the mortality rate, extended the development duration, and decreased the emergence rate of adults. I suggest accepting this manuscript after a minor revision. Here are some specific suggestions.
Line 2: Laccase2 or Laccase 2
Line 158:“BlasP” should be “BlastP”
Line 191 the letter p should be upper-case and italic?
Delete line 206-208. “The expression of BtLac2 gene was detected by RT-qPCR at 72 h after treatment of B. tabaci nymphs with the mixture of dsRNA and nanomaterial.”
Line 235 dsBtLac2-droped use dsBtLac2-treated?
Line 245 have a combined effects on? The word definitely has an absolutist tone.
Line 298 B. tabaci not B.tabaci
Line 317 were not was.
Line 338 Bemisia tabaci should not abbreviated when in the start of one sentence.
Line 314-331 in the discussion part the authors should provide some examples of laccase 2 function in other insects from published studies.
Author Response
Dear reviewer,
Thanks a lot for the constructive suggestions. We have carefully revised the manuscript point by point. Please see the attachment.
Best regards,
Chunhong

Reviewer 2 Report
The manuscript entitled 'Involvement of Laccase2 in cuticle sclerotization of the whitefly, Bemisia tabaci Middle East-Asia Minor 1' is a well-written article about whitefly Laccase2 characterization and importance in cuticle sclerotization.
However, I do have some questions regarding the methods.
Is there any other reference where dropping dsRNA on the back worked? Did the authors try microinjection?
How did the authors decide the time-point (72hrs) for gene expression studies? Was the gene-expression studied after 1d or 2d post dsRNA exposure?
I would suggest the authors put an illustration or a picture describing the dsRNA treatment.
It is always good to have a control with mock treatment along with eGFP control, to show that eGFP doesn't cause any difference. I suggest the authors use a mock control using just water for the measurement of cuticle and phenotype.
Measurement of cuticle is not properly explained in the methods. Add a few lines as to explain how it was done or what software was used to measure.
Author Response

(The authors gave the same response as above.)

Reviewer 3 Report
Comments and suggestions are in the attachment.

Author Response
Response to Reviewer 3 Comments
Point 1: Line 30: Cancel “further”
Response 1: Done.
Point 2: Line 48: Change “formation” to “synthesis occurs”
Response 2: Done.
Point 3: Line 58: Change “diseases” to “viruses”.
Response 3: Done.
Point 4: Line 60: Add “also known as B biotype” .
Response 4: Done.
Point 5: How often the purity was checked once a month or once every two months?
Response 5: Done.
Every two months.
Point 6: How many whiteflies were used for total RNA extraction?
Response 6: Done.
Approximately 200.
Point 7: Line 83: Add “Respectively”
Response 7: Done.
Point 8: provide the sequence of the gene region used for dsRNA syhthesis
Response 8: Done.
Please see the Supplementary Materials Figure S2.
Point 9: Line 120: Change “before” to “described above”
Response 9: Done.
Point 10: Line 127 “72h of ...treatment”
Response 10: Done.
Point 11: Line 135: Cancel “mounted”
Response 11: Done.
Point 12: Line 154: Change “was” to “is”
Response 12: Done.
Point 13: What kind of phylogenetic analysis was conducted? NJ, Maximum likelihood or Bayesian? provide more details on it.
Response 13: Done.
The phylogenetic tree was generated by MEGA 7.0 based on the neighbour-joining method.
Point 14: Details on experiments conducted to detect gene expression in different life
stages is lacking in material and method section.
Response 14: Done
RNA was isolated from different developmental stages (eggs, 1st instar nymph, 2nd instar nymph, 3rd instar nymph, 4th instar nymph, and adults). Approximately 200 individuals of B. tabaci were used for one biological replicate. Adults that emerged within two days were used in the experiment.
Point 15: Line 243: Cancel “present”.
Response 15: Done
Point 16: Line 317: Change “was” to “have been”
Response 16: Done.
Point 17: Line 333: May be because they cannot transmit the virus
Response 17: Done.
Add “or may be because they cannot transmit the virus” after the sentence “Due to the limited feeding capacity“

Round 2
Reviewer 3 Report
The reviewer went through the revised version of the manuscript, "Involvement of Laccase2 in cuticle sclerotization of the whitefly, Bemisia tabaci Middle East-Asia Minor 1". And found that all comments and suggestions have been addressed and the manuscript is presented as per journal guidelines. Therefore, the manuscript may be considered for publication in the journal Insects.